# Extensive Ruminant Production Systems and Milk Quality with Emphasis on Unsaturated Fatty Acids, Volatile Compounds, Antioxidant Protection Degree and Phenol Content

**DOI:** 10.3390/ani9100771

**Published:** 2019-10-08

**Authors:** Andrea Cabiddu, Claudia Delgadillo-Puga, Mauro Decandia, Giovanni Molle

**Affiliations:** 1Agris—Servizio Ricerca per la Zootecnia, Loc. Bonassai, 07040 Olmedo, Sassari, Italy; mdecandia@agrisricerca.it (M.D.); gmolle@agrisricerca.it (G.M.); 2Departamento de Nutrición Animal, Instituto Nacional de Ciencias Médicas y Nutrición Salvador Zubirán (INCMNSZ), CDMX 14080, Mexico

**Keywords:** dairy products quality, grazing ruminants, pasture, volatile compounds, fatty acids, phenols

## Abstract

**Simple Summary:**

This paper updates the knowledge on the effects of grazing ruminants on milk quality and cheese with emphasis on unsaturated fatty acids, volatile compounds antioxidant protection degree and phenols. It focuses on the effects of the forage species and its phenological phase on the fatty acid (FA) profile of the forage and the milk/cheese fatty acid profile. In addition, this paper highlights that milk and cheese sourced from grazed herbage is characterized by a higher content of volatile compounds compared to cheese made from sheep fed at stall. The volatile compounds, besides giving a characteristic flavor to the cheese, can also be used as biomarkers because they can be transferred from herbage to the milk. Recent results show that some endogenous plants factors are capable, when properly included into ruminant’s diet, to modulate feed digestion and nutrient uptake, making livestock systems more efficient and environmentally sustainable. Finally, of particular interest is the role of grazing ruminants in land management and landscape re-evaluation for tourism purpose, a key element to prevent the depopulation and degradation of rural areas.

**Abstract:**

Dairy products from grazing ruminant have numerous positive effects on human health thanks to their higher content essential fatty acids, vitamins, and polyphenols. Compared to livestock fed a conventional maize silage- and/or grain-based diet, grass-fed livestock produce milk with higher levels of n-3 fatty acids, vitamins A, E, carotenoids, and phenols. The effect is even more pronounced if animals are grazing on legume/forbs-rich grasslands. This review argues, based on the available literature, about the effect of grazing ruminant on milk and cheese quality, including the hedonistic aspects, pointing out the link between territory and dairy products quality (Protected Designation Origin; Protected Geografic Origin; namely PDO and PGI labels). Moreover, it points out the main plant biomarkers which can be used to discriminate grazing sourced from stall-fed sourced milk and dairy products. Overall milk and cheese sourced from grazing animals (cows, sheep and goat) showed higher levels (compared to stall system) of FA, vitamins, phenols, putatively beneficial for consumers’ health. FA and plant secondary metabolites can also affect flavor and some nutritional and technological features of dairy products such as their antioxidant protection degree. This would favour a fair pricing of dairy products sourced from grazing systems and the persistence of viable and sustainable extensive production systems.

## 1. Introduction

The Academy of Science for enhancing the health of adults recommends a food-based approach through a diet that includes regular consumption of fatty fish, nuts and seeds, lean meats and poultry, low-fat dairy products, vegetables, fruits, whole grains, and legumes [1]. These recommendations are made within the context of rapidly evolving science delineating the influence of dietary fat and specific fatty acids (FA) on human health, likewise saturated fatty acids (SFA). The principal dietary sources of SFA in most Western countries are full-fat dairy products and red meat, the former also being rich in trans FA (TFA) and low in polyunsaturated fatty acids (PUFA) of the n-3 series [2].

In particular, dairy products provide 25%–60% of the overall saturated fat consumption in Europe, which makes them, since decades ago, a target of dieticians’ criticism due to the negative effects of excessive consumption of saturated FA on human health [3]. Ruminant product fats are relatively more saturated than most plant oils and this is also a consequence of biohydrogenation of dietary polyunsaturated fatty acids (PUFA) by rumen bacteria (Figure 1). 

As a consequence of this imbalance, a great effort has been made by researchers to enhance ruminant products quality, in particular focusing on the influence of dietary forages on fatty acid composition of milk and dairy products [5,6,7,8,9]. Products from ruminants also contain other minor components which are deemed able to improve health and wellness of consumers, like vitamins A and E, phenols and other secondary compounds like volatile compounds with a special role for the hedonic aspect. Actually, several reviews reported that, in ruminants, grazing gives the best chance to achieve this target compared with stall feeding, since it combines better product quality with low feeding costs [9,10,11]. Moreover, pasture-based systems are usually perceived by consumers as more environmentally and animal-welfare friendly than housed systems. In addition, milk from ruminants grazing legumes and other forb-based pastures usually contain phenols. It is known that many of the claimed health benefits in human diets have been associated with the phenolic compounds (e.g., resveratrol in wine, [12]), hence, the occurrence of some phenolic compounds in dairy products could give more added value to products obtained from grazing ruminants [10]. Finally, other work has highlighted the positive effect of grazing on product technological and organoleptic attributes such as texture, oxidative stability and flavor. This paper is aimed at reviewing the above aspects and suggesting the best up-to-date bio-markers to trace grazing diets of ruminants up to their products, with particular reference to milk and dairy products.

## 2. Influence of Pasture on Milk and Cheese Quality

### 2.1. Cows

#### 2.1.1. Botanical Composition of Pasture and Milk Fatty Acid (FA) Profile

The relationship between animal feeding and the characteristics of several quality identified products (protected designation of origin (PDO), protected geographic indication (PGI)) is increasingly perceived as important by consumers. Moreover, a strong link between quality and the livestock production system is requested for these products, as reported by Grappin et al. [13]. This author showed a clear relationship between edaphic and flora characteristics of the production areas and the aroma of the different cheeses (Figure 2). For example, in the Comté cheese area, they discriminate nine homogenous areas linking typical taste of cheese and geological area where farms were located with a large number of forage species detected (about 426). 

Furthermore, in grazing cows, Bugaud et al. [14] showed that the botanical composition of pastures and native meadows can influence the fatty acid profile of milk and cheese (Figure 3).

Fatty acid composition in grass changes among forage species, variety/cultivar, regrowth stage, phenological stage, leaf/stem ratio, light intensity, nitrogen fertilization level and other pasture management techniques, as reported by Elgersma [9]. Milk from cows grazing a native pasture with higher botanical biodiversity contains more PUFA, less saturated FA and is featured by a lower n-6:n-3 ratio than milk sourced from cows fed at stall [15,16,17], with an associated increase of conjugated linoleic acid (CLA) by 300%. Linseed and other oilseeds are also very rich in PUFA precursors and their supplementation can enhance some beneficial FA in dairy products. However, the use of these supplements can increase feeding costs and could markedly reduce the reliability of the discrimination between grazing and stall-feeding systems based on milk fatty acid profile only. 

Cheese and butter texture (e.g., spreadability) can also be influenced by fatty acid profile, as reported by Martin et al. [18]. In addition, cows grazing in nitrophilic meadows (*Rumex alpinus* prevalence) gave a cheese crumblier than that sourced from cows grazing in lowland damp meadows (*Dactylis glomerata* prevalence), probably because the PUFA level is higher in fat from highland than lowland dairy products. In alpine conditions, cows grazing *Trifolium alpinum*-dominated pastures gave a milk richer in long chain FAs, unsaturated and monounsaturated FAs, and odd- and branched-chain FAs (OBCFA) while those from *Festuca nigrescens* pastures contained more short- and medium-chain FAs, SFA, and α-linolenic acid (ALA) as a probable consequence not only of differences in levels of FA precursors but also of plant secondary metabolites (PSM) [19]. These results are in partial agreement with Tornambé et al. [20], who found that pastures with different ratio of gramineous/dicotyledons and hence with a different content of aromatic precursors species, affect the linoleic acid (LA) and ALA level in milk. In particular, Coppa et al. [21] underlined this effect, explaining it by the presence of plants belonging to *Apiaceae* and *Geraniaceae* families, which often occur in meadows. Forbs are plants rich in PSM that could inhibit rumen biohydrogenation of PUFA. Seasonal effect (different times of the biological cycle) is also important: from May to July the contribution of the *Poaceae* family decreases while the contribution of *Dicotyledon* families increases in Alpine grassland. In this situation, an increase of SFA and a decrease of PUFA, as a result of the change of the phenological stage of plants, is reported by Grappin and Coulon [13]. This paper also points out the importance of the phenological stage of plants during the grazing period, highlighting that the optimum FA profile usually overlaps early growth. In addition, Gorlier et al. [22] found a positive relationship between *Cyperaceae* and *Fabaceae* contribution on native pasture in the Alpine region and level of PUFA and MUFA in milk. Actually, results from eight forage species from the grassland grown at 650 m a.s.l., cut at two regrowth periods, showed that LA and ALA decreased with the age of the plant, with higher levels of PUFA in the legume family [23]. Under controlled conditions, in cows feeding different fresh forage species, Kälber et al. [24] found that the milk sourced from beerseem clover (*Trifolium alexandrinum,* var. Sacromonte) gave the highest level of milk PUFA and n-3 FA while that from ryegrass (*Lolium multiflorum* ssp. Westerwoldicum, var. Saproso) the lowest. Moreover, as summarized by Moloney et al. [8], Alpine pastures include legumes and forbs species rich in PSM, which play an important role on PUFA rumen biohydrogenation (Figure 1). In fact, pastures which contain berseem clover, white clover, red clover, birdsfoot trefoil or sulla are able to increase linolenic acid in milk [8]. Apparently, similar effects on milk and cheese composition are found in the Mediterranean basin when we observe cows grazing native pastures rich in *Asteraceae* and *Geraniaceae,* which showed higher levels of PUFA compared to milk sourced from animals fed at stall [25]. These results are confirmed by Bonanno et al. [26] who found higher levels of PUFA, MUFA and n-3 FA in cheese from grazing than stall-fed cows. 

#### 2.1.2. Minor Pasture Components and Cheese Flavor

Grassland forages in interaction with local environmental factors (pedology and clima) can influence the sensory properties of milk. These effects may result directly from compounds originating from forage (carotenoids, terpenes and other PSM) or indirectly through forage-related changes in animal physiology. Primary results were sourced from a study in the Alpine areas comparing different grassland paddocks with different predominant plants [25]. Results from this study underlined that Abondance cheeses sourced by multi-species pastures rich in grasses (comprising predominantly *Agrostis capillaris* and *Nardus stricta*), associated with 32 other species were saltier, more bitter and had an aroma more intensely characterised by sour, burned, toasted and fermented vegetables than cheese from pastures rich mainly in grasses such as *Festuca rubra.* In addition, Buchin et al. [27] supposed that *Ranunculus* spp. could influence the different levels of proteolysis by changes of plasmin level in milk and in cheese, hence explaining the difference in bitterness found between the cheeses (Figure 4). 

Similar results were obtained comparing milk from pastures rich in *Apiaceae*, *Asteraceae* and *Plantaginaceae*, which show high levels of terpenes with milk from pastures rich in *Gramineae* [14]. Despite the large amount of volatile compounds found in milk (totally 262 compounds) only 12 monoterpenes were significantly correlated with the monoterpenes found in pastures (R^2^ = 0.64, *p* < 0.01). Pasture richness in the diet could also enhance the oxidative stability of milk, as observed by Martin et al. [15], with higher levels in milk of lutein, carotene, vitamin A and E than stall-fed cows. These results are not confirmed by Kälber et al. [24] and Tornambé et al. [28] who did not find any correlation between botanical composition of grassland and milk content of carotene, vitamin A and E.

A global approach to better understand the effect of botanical composition on milk and cheese microcomponents was adopted by Martin et al. [18], who pointed out the effect of the *Poaceae* and *Leguminosae* families on propionic and butyric acid content in Abondance cheeses and its pungent flavor while *Ranuncolaceae* and *Rosaceae* are more responsible for fruity and animal flavor in cheese. The main groups of volatile compounds found in milk belong to monoterpenes and sesquiterpenes class, depending on the chemical composition of pasture [29]. 

#### 2.1.3. Minor Pasture Components and Cheese Texture

Bugaud et al. [14] in a trial conducted in two farms producing Abondance cheese, showed wide deviations in cheese texture within the same farm, depending on grazed meadow characteristics. Although the widest deviation was observed between valley and highland pastures, there was also a degree of variability within the same highland meadow. The main differences involved cheese texture that was more cohesive, elastic and deformable in valley than in mountain cheeses, and crumblier in cheeses from nitrophilic and snowbound paddocks than in cheeses from damp meadows.

#### 2.1.4. Antioxidant Activity and Phenols Contents in Milk and Cheese

A recent paper with a holistic approach (including economic, social, livestock management and environmental variables) underlined the role of grazing dairy cow system on aromatic volatile compounds that may impact on sensory perception [30]. Between the differents milk volatile compounds, p-cresol appear a very important attribute for milk flavor, furthermore, p-cresol to belonge to phenol components, derived from rumen metabolism of isoflavones present in clover pasture. Actually, phenols components which occurred in plants appear as very interesting molecules thanks to their antioxidant, antimutagenic and antitoxic capacities reported by Veskoukis et al. [31], following the results from in vitro, in cell lines and in vivo experiment.

### 2.2. Sheep and Goat

#### 2.2.1. Botanical Composition of Pasture and Milk Fatty Acid Profile

Grazing of dairy sheep in the Mediterranean basin is mainly based on extensive and semi-extensive grazing systems. Grasslands in the Mediterranean basin show a greater extent of biodiversity compared with other grasslands of central Europe [28,29,32,33]. This is of interest to build a strategy to increase the added value to dairy sheep and goat products sourced from livestock systems with putatively low environmental impact. The first review on the effect of Mediteranean pasture botanical composition on sheep milk FA profile [34] pointed out that forage species and phenological stage of plants are a key factors of milk fatty acid composition. Higher levels of PUFA precursor were found in legumes (*Trifolium subterraneum*, *Hedysarum coronaryum* SU, *Medicago polimorpha* BM) compared to *Lolium rigidum* (RY) and *Chrysanthemum coronarium* (CH) which reflect the higher content of PUFA in milk. Comparing such forage species freshly cut and offered as the only feed to dairy sheep at stall we observed the following ranking for PUFA SU > BM > CH > RY during vegetative stage, while during reproductive stage the rank was CH > BM > SU > RY [35], Figure 5. 

These authors concluded that forage species can strongly affect the milk FA profile with different magnitude, in terms of short, medium, and long chain fatty acid, including CLA. The level of PUFA in forages is not the only factor which affects milk fatty acid profile: during the flowering stage, we observed that although sulla contains less linolenic acid than burr medic, the level of linolenic acid is higher in milk from sulla than burr medic. This is probably due to the content of tannins in sulla which decreases the ruminal biohydrogenation of linolenic acid by about 30%, resulting in an increase of linolenic acid content in milk [36] (Figure 6). 

Usupplemented dairy sheep grazing a mixed pasture (*Trifolium repens*, *Lolium perenne*, *Medicago sativa*, *Festuca arundinacea* and *Trifolium fragiferum*) had a higher content in milk of MUFA, PUFA, CLA, BCFA, OCFA compared to sheep supplemented daily with 600 g/ewe of corn grain with different botanical composition. The effect of different forage species in sheep milk is also confirmed in a survey carried out by Cadiddu et al. [37], who found higher levels of CLA (100% more, *p* < 0.05) and linolenic acid (100% more, *p* < 0.05) in sheep grazing legumes than grasses. These results are in partial agreement with data by Addis et al. [38], who reported a two-year survey on 40 dairy sheep farms in Sardinia. These authors found no clear relationship between legumes in the grassland and levels of milk PUFA. The importance of pasture on milk fatty acid profile is also clearly shown by Biondi et al. [39] who found, after sudden passage of Comisana sheep from stall-feeding to only pasture with vetch and oat, strong changes of FA content in milk: −10% of SFA, *p* < 0.01; +13% of MUFA; +70% of PUFA (*p* < 0.01); +320% of n-3:n-6 ratio; +170% of CLA. In the Alpine area, comparing different grassland at different altitude milk fatty acid composition of dairy sheep changed with a minor magnitude than that observed in the Mediterranean area. Grassland located in mountain (1500 m.a.s.l.) areas sourced milk with higher n-3 FA (+17% *p* < 0.01), PUFA (+10% *p* < 0.001) and CLA (+30% *p* < 0.01) than grassland located in the valley (500 m a.s.l.) [40]. Comparing different pasture mixtures (annual ryegrass + burr medic (RY + BM); annual ryegrass + sulla (RY + SU) or annual ryegrass + subclover (RY + SC)) [35] did not observe changes between mixtures on CLA, vaccenic and linolenic acid content in sheep milk, despite the high proportion (40%) of legumes in RY + SU and RY + SC treatments. This is probably due to the high quality of ryegrass, which showed the highest level of linolenic acid in herbage samples. Moreover, the authors observed a very interesting effect from leafiness of forages correlated with the level of vaccenic acid and estimated Δ9 desacturase activity [41]. On the other hand, the same research group [42] compared a ternary mixture including *Chrysanthemum coronarium* L., *Lolium rigidum* Gaudin and *Medicago polymorfa* L. with a binary mixture based on the same species, except the *C. coronarium* (control pasture). The authors observed that grazing the mixture with *C. coronarium* increased the level of vaccenic acid and CLA (+58%, *p* < 0.01) in sheep milk, compared to the control group [42], (Figure 7).

Milk from sheep grazing pure pastures of safflower (*Carthamus tinctorius*, SA), chicory (*Cichorium intybus*, CH), or burr medic (*Medicago polymorpha*, MP) showed the following ranking for the level of CLA: 20.69, 15.98 and 15.17 mg/g of fat methyl esters (FAME) for SA, MP and CH, respectively (*p* < 0.01) [43]. A recent paper suggests that oilseed-based supplementation in grazing dairy sheep needs to be evaluated more carefully, not only in terms of the enrichment of milk PUFA, CLA *cis*-9, *trans*-11 and vaccenic acid, but also considering the increase of some milk trans fatty acids which may be deleterious for human health [44]. In contrast, with oilseed supplementation, feeding only fresh forages at pasture, when possible, seems a win-win option because it combines better product quality, with low feeding costs and usually brings about positive consumer perceptions. Sheep grazing pure grass (*Lolium multiflorum* Lam) without supplementation outperformed for the n-3:n-6 ratio oilseed-supplemented sheep, showing also a lower level of trans fatty acids [44], as shown in Table 1.

Grazing can also enhance other interesting minor components belonging to the antioxidant compounds, included vitamins A, E and antioxidants, cumulatively evaluated using the antioxidant protection degree (DAP). Interesting results from goat milk [10,45] underlined the role of grazing systems to increase this index in milk and cheese compared to animals fed at stall. It is noteworthy that DAP was two-fold higher in cheese from goats fed only fresh herbage than supplemented goats. Similarly [45], DAP values were found 10 times higher in grazing goats than stall-fed goats. In addition, recent evidence demonstrates that goats reared in shurbland can increase PUFA, DAP and phenols content in milk compared to animals fed at stall [10,46], as shown in Table 2 and Figure 8. 

#### 2.2.2. Minor Pasture Components and Cheese Flavor

As reported for fatty acid profile, the inclusion of *Chysanthemum coronarium* in a grazing sheep diet can impact on the sensorial profile of cheese [48]. This daisy plant was shown to contain several volatile compounds, the main ones are: 1-methyl-4-(1-methylethylidene)-ciclohexene (terpinolene, 47%) and ethanol (36%). Among other volatile substances, terpenes such as a-pinene, triciclene and camphene, cyclic and unsaturated hydrocarbons and esters were detected. The inclusion of *C. coronarium* in sheep diet affected the composition of volatile fraction of milk and cheese. In particular, the milk and cheese produced from sheep grazing the ternary mixed pasture with the daisy plant, unlike the one sourced from control grass-legume pasture, was featured by terpenes such as terpinolene, triciclene, 3,7- dimethyl-1,6-octadiene, α-pinene and camphene. A panel-test based on olfactory evaluation showed that the cheese obtained from sheep grazing the ternary mixture was distinguishable from that of their counterparts grazing the grass-legume mixture. 

A study on goats grazing a native pasture revealed that volatile compounds in milk and cheese changed in quantity and type with the progressing of seasons and for this reason, the fragrance profile of products also changed. Moreover, it appeared that in winter and spring, terpenes were not the most important class of volatile organic compounds (VOC), whereas they reached higher levels in summer [49]. These results agree with Decandia et al. [50], who compared the effect of goats browsing a Mediterranean lentisk-based shrubland with stall-fed goats: the milk from the browsing goats was higher in CLA and VOC (particularly ketones and aldehydes) than the milk from the goats fed at stall (Figure 9). However, chicory-based grassland could affect the bitterness of cheese, because of the presence of PSM (mainly sesquiterpenes) in this forage [51]. 

### 2.3. Plant Biodiversity Secondary Metabolites and Their Role on Healthiness and Sensorial Features of Animal Products

Plants contain secondary metabolites (PSM) that interact in a variety of ways. PSM and its regulation is strongly susceptible to environmental influences and to potential herbal predators, abiotic and biotic factors which might be specifically induced by means of various mechanisms, creating a wide variation in the biogenesis and accumulation of these metabolites. Toxic effects of plants vary depending on the level and type of PSM. The toxicity of PSM and respective concentration shifts may have severe consequences on livestock production and health. Such PSM are also of interest in livestock production because when their concentration in the diet is low, they can positively influence animal performance and quality of products. The main PSM of our interest are: tannins (phenols), polyphenol oxidase and coronaric acid. Tannins are a heterogeneous group of PSM known for both beneficial and detrimental effects on digestive physiology. Tannins supplementation of ruminant diets alters both in vivo and in vitro unsaturated fatty acids biohydrogenation, hence the profile of fatty acids outflowing the rumen, which can influence milk and meat content of beneficial fatty acids such as linolenic acid (*cis*-9, *cis*-12, *cis*-15-18:3), vaccenic acid (*trans*-11-18:1) and rumenic acid (*cis*-9, *trans*-11-18:2), among others. Published information indicates that tannins could inhibit biohydrogenation affecting ruminal microorganisms. Some studies [52] found increments in linolenic, rumenic and/or vaccenic acids in meat and milk fat, supplementing ruminants with different sources of tannins, however, the effects of tannins supplementation on milk and meat fatty acid profile are not consistent [6,22]. 

On the other hand, the polyphenol oxidases (PPO) are a group of copper metalloenzymes which include: catecholase (EC 1.10.3.2), laccase (EC 1.10.3.1), and cresolase (EC 1.14.18.1). Catecholase is the most dominant polyphenol oxidases in forage crops and will be referred to hereafter as PPO [53]. 

Polyphenol oxidase catalyze the oxidation of ortho(o)-phenols to o-quinones at the expense of molecular O_2_. The reaction products, o-quinones, are highly reactive electrophilic molecules which act to covalently modify and cross-link a variety of cellular nucleophiles including quinone-quinone self-polymerization. The formation of o-quinone adducts that result in browning of various plant tissues represents the detrimental effect of PPO in post-harvest physiology and food processing and is one of the primary reasons for interest in this enzyme. The mechanism for protecting glycerol-based PUFA has yet to be fully elucidated but may be associated with entrapment within the protein-bound phenols (PBP) reducing access to microbial lipases or differences in rumen digestion kinetics of the forage and therefore not related to PPO activity. Overall, the reduction effect of PPO on the PUFA biohydrogenation is about 10% compared to control diet, leading to an increase of beneficial fatty acid content in milk and meat. Recently [54], it was shown that native plants from the Mediterranean area have high values of PPO, in particular *Medicago arabica* and *Trifolium resupinatum*. This is of interest to explain how milk and meat fatty acid composition could be modulated by the inclusion of different forage species. 

### 2.4. Antioxidant Activity and Phenols Contents in Milk and Cheese

As reported above, in the Mediterranean basin, the small ruminant (dairy sheep and goat) livestock system is mainly based on a grazing system. At the moment, the references on milk and cheese antioxidant activity are mainly based on Vitamin E detection, while not enough data are available on milk and cheese phenols content. Recently, an interesting paper [10,47] has shown that when comparing goat stall feeding with the grazing system, the level of phenols increases dramatically in the grazing system. These results are expected since shrubland, brush and natural pasture are rich in phenolic compunds. 

## 3. Traceability and Authenticity of Ruminant Products

Council Regulation (EEC) no. 2081/92 (1992), attempts to unify different concepts of typicity. “Typicity” is indeed part of the Designation of Origin (PDO, Protected Designation of Origin, and PGI, Protected Geographic Indication, labels) concept developed in the EU with the aim of gaining consumers’ confidence towards products of specific European regions. The Council Regulation (EEC) no. 2081/92 (1992) lays down rules on the protection of agricultural products intended for human consumption. It distinguishes between two categories of protected designations: Protected Geographical Indications and Protected Designation of Origin. These categories differ depending on the nature and the intensity of the link between the product and the defined geographical area, usually quoted on the product’s label. The Protected Designation of Origin concerns products very closely associated with the region where they are produced, processed according to a well-defined technological scheme. In the years 1990–1995 after the bovine spongiform encephalopathy (BSE) and Foot and Mouth Disease outbreaks, which arose from the use of contaminated animal-based feedstuffs, European consumers increased their demand for transparent and clear information on the livestock production system implemented to deliver the food available in the EU markets. It is well recognized that grassland has added value of milk and meat [55], but this needs to be quantified to discriminate between products from grassland and other products. Actually, there is an increasing interest from consumers for clear information on health benefits associated with consumption of food and/or the green image of animal products. All of these factors have also led to the introduction of an increasing number of quality assurance schemes in the agrifood sector [55]. Traceability and authenticity are well defined by Prache et al. [56] and Coppa et al. [57]. The traceability is defined as “the ability to follow the movement of a food through specified stages of production, processing and distribution”. Authenticity is defined “the process by which a food is verified as complying with its label description”. In particular, it is of interest to quantify the effect of levels of pasture inclusion on animal diet, characterizing milk and meat composition/quality. As reported by Prache et al. [56], several approaches are available to detect biomarkers or markers of pasture in animal products. We can distinguish four main strategies [52]: three are based on the determination of markers: plant biomarkers (A), metabolic markers (B), and physical markers (C). The fourth is the global approach (D). The following chapter will focus in particular on plant biomarkers. For the other strategies, readers can refer to other recent reviews [57].

## 4. Plant Biomarkers

### 4.1. Fatty Acids

Geographical authentication of sheep milk is an issue for the production of PDO-labelled cheeses. With the aim to set up a fast methodology for tracing PDO cheeses obtained in different areas of Sardinia, two hundred and fifty samples of sheep milk from different areas of Sardinia were scanned by Mid-Infra-Red (MIR) reflectance spectroscopy. The analysis of spectra was submitted to genetic algorithms with references of FA to select the informative variables for developing discriminant models able to correctly classify the samples. This model was validated obtaining 96% of the correct prediction [58]. This approach was also developed for cow milk at the European level to predict diet composition of animals. In particular, the prediction based on FA profile was excellent for the proportion of fresh herbage in the diet (R^2^ = 0.81) [59]. Similar results were obtained to classify sheep and beef meat origin (grazing versus stall) on the basis of FA profile, particularly with linolenic and C20:5 FA [60].

### 4.2. Carotenoids

Carotenoids are natural pigments occurring in plants: lutein is the only carotenoid found in the fat tissue of sheep, whereas cattle also store carotenes in their fat depots [61]. Content of carotenes and carotenoids in forages decrease with age of plant and with phenological stage and with conservation method (fresh versus silage, hay or dehydratation). Comparing grazing sheep on pasture at two different phenological stages, we observed a decrease of vitamins A and E passing from the vegetative to the reproductive stage of forage [62]. Pasture species-richness in the diet can also enhance the oxidative stability of milk [63], with higher levels in milk of lutein, carotenes, vitamins A and E than in milk obtained from stall-fed cows.

### 4.3. Terpens

Most of the volatile compounds in milk are originated by deep changes of feed components during digestion and intermediate metabolism due to microbial and enzymatic processes up to mammary synthesis. Some volatile agents, such as monoterpenes and sesquiterpenes, are putatively able to pass from the feed to the dairy products without modification [63,64,65]. Other studies showed, however, that a partial degradation of terpenic compounds by the rumen fermentation could happen [25,66]. Some authors evidenced that the content of terpenes in the pastures is related to their botanical composition. Gramineae resulted poorer in terpenes than *Asteraceae*, *Apiaceae* and *Plantaginaceae* [67,68]. 

### 4.4. Phenols

Milk phenols content could also be of interest as biomarkers especially in a goat extensive livestock system. Results from a recent survey carried out in Sardinia showed that goat browsing Mediterranean maquis (shrubland) mainly based on *Pistacia lentiscus* L., *Quercus ilex* L., *Arbutus unedo* L. *Cistus monspeliensis* L. and *Cistus incanus* L., showed a higher level of milk phenols compared to milk from goats raised under confined conditions and stall-fed concentrate and hay [45]. 

### 4.5. Redox Biomarkers

The evaluation of redox activity includes all the activity which study the dynamics in reinforcing the antioxidant and antitoxic defence of tissues and/or whole organisms to prevent the activity of free radical-related disease onset [31]. Redox biomarkers represent the activity of different microcomponents which occur in milk like usaturated fatty acid, phenols and others pigments and metals like iron. As reported recently by Delgadillo-Puga et al [10], animal grazing in natural pastures increase the level of 1,1-diphenyl-2-picrylhydrazyl (DPPH), ferulic acid and catechin in milk compared to the conventional diet. DPPH is a methodology used to test antiradical properties, and their activity in milk is linked to catechin and ferulic acid content (phenols) in milk. So, the level of phenolic compounds in the ruminant diets can increase the redox power of milk and cheese.

## 5. Conclusions

Milk and dairy products of ruminants reared at pasture are usually rich in FA, vitamins, polyphenols and related enzymes (PPO), putatively beneficial for consumers’ health. The level of these compounds depends upon the level of their precursors in ruminant diets but also on the level of PSM which can modulate their metabolism and uptake from the gastro-intestinal tract. FA and PSM can also affect flavor and some nutritional and technological features of dairy products such as their antioxidant protection degree. Methods have been developed to trace the ruminant diet based on pasture up to the ultimate products. Some of them provide encouraging results for discriminating at a low cost, dairy products based on pasture. This would favour a fair pricing of dairy products sourced from grazing systems and the persistence of viable and sustainable extensive production systems. 

## Figures and Tables

**Figure 1 animals-09-00771-f001:**
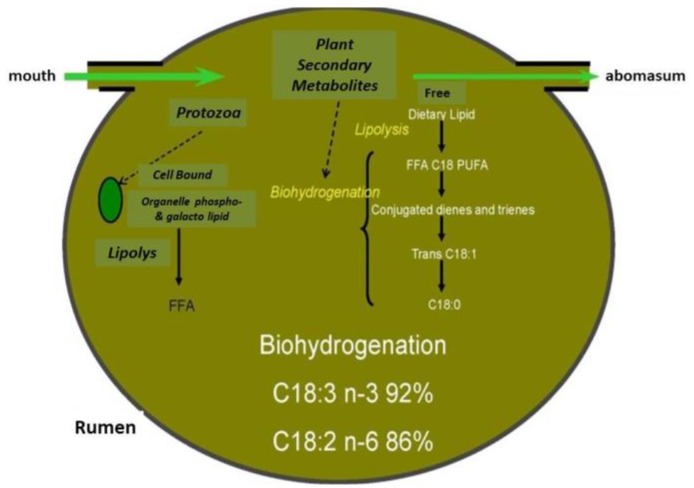
Diagrammatic representation of lipid metabolism in the rumen including the action of plant secondary metabolites [4].

**Figure 2 animals-09-00771-f002:**
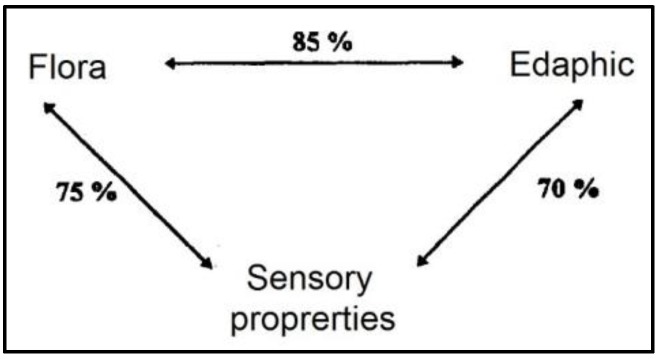
Relationship between edaphic flora characteristic and aroma profile of cheese [13].

**Figure 3 animals-09-00771-f003:**
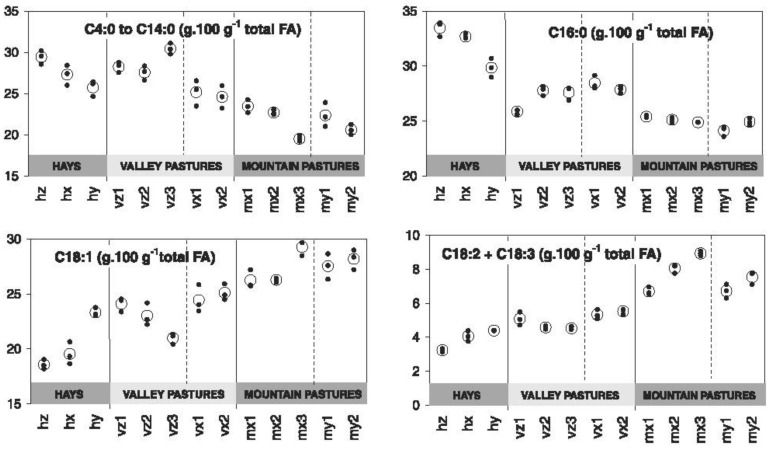
Effect of botanical composition on milk fatty acid (FA) composition of cows. Legend: x, y, z = farms, h = hay, v = valley pasture, m = mountain pasture, 1–3 = date of sampling [14].

**Figure 4 animals-09-00771-f004:**
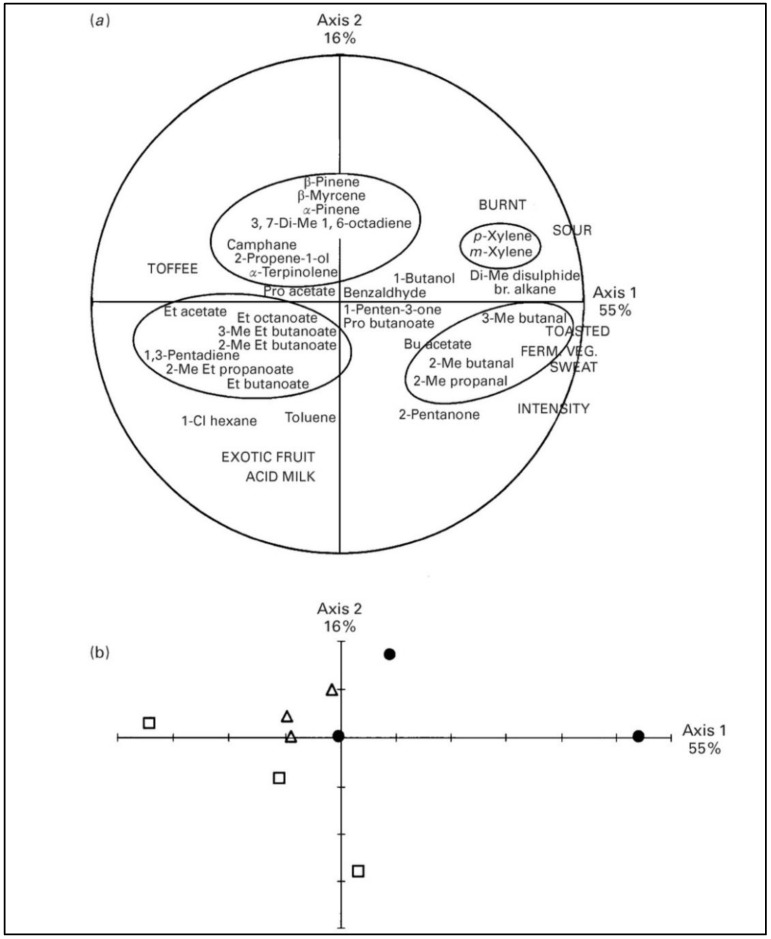
Principal component analysis (PCA) of the aroma characteristics of the cheeses: plot of principal axes 1 and 2. (**a**) Correlation circle. Aromas are represented in capitals. Volatile compounds that varied significantly (*p* < 0.05) between series of cheeses by analysis of variance are added as additional variables. Compounds of the same chemical family, characteristic of a series of cheeses, are included in ovals. FERM. VEG. = fermented vegetable; Bu = butyl; Cl = chloro; Et = ethyl; Me = methyl; Pro = propyl; br = branched. (**b**) Representation of the experimental cheeses from north and south areas in the study: ∆ = south 1 cheeses; □ = south 2 cheeses; ● = north side cheeses [27].

**Figure 5 animals-09-00771-f005:**
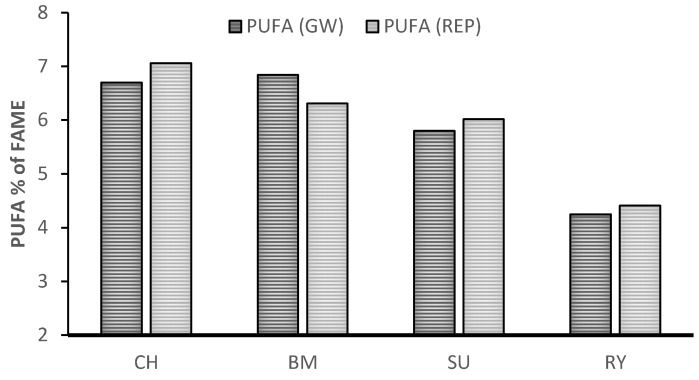
Level of milk polyunsaturated fatty acids (PUFA) of sheep feed with fresh herbage cut at vegetative (GW) and reproductive stage (REP). Annual ryegrass (RY, *Lolium rigidum* Gaudin), sulla (SU, *Hedysarum coronarium* L.), burr medic (BM, *Medicago polymorpha* L.), and a daisy forb (CH, *Chrysanthemum coronarium* L.) [35].

**Figure 6 animals-09-00771-f006:**
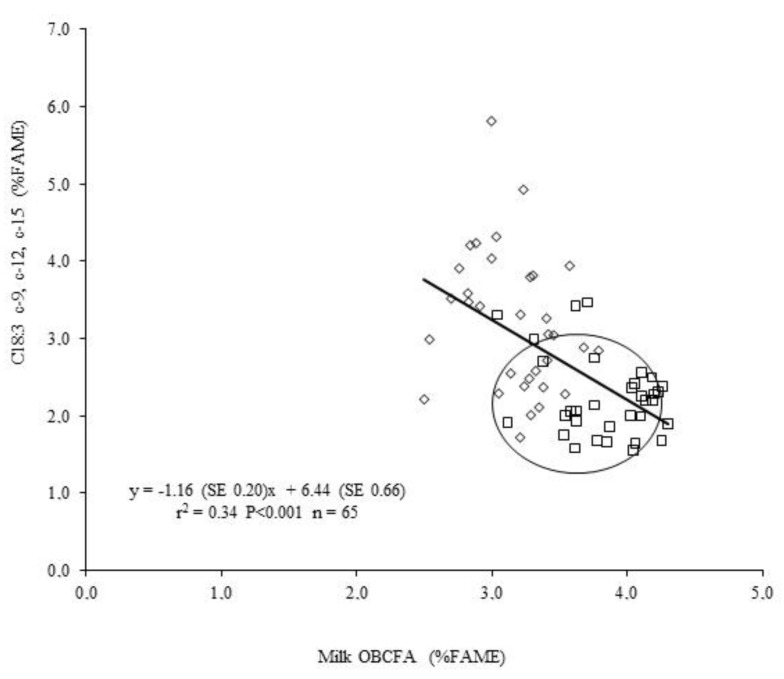
Relationship between odd- and branced-chain fatty acid (OBCFA) and C18:3 c-9, c-12 c-15 content in milk of sheep grazing sulla either receiving or not receiving an antitannic supplementation (each point is an individual value: ◊ = sulla with tannin effect; □ = sulla without tannin effect) [36].

**Figure 7 animals-09-00771-f007:**
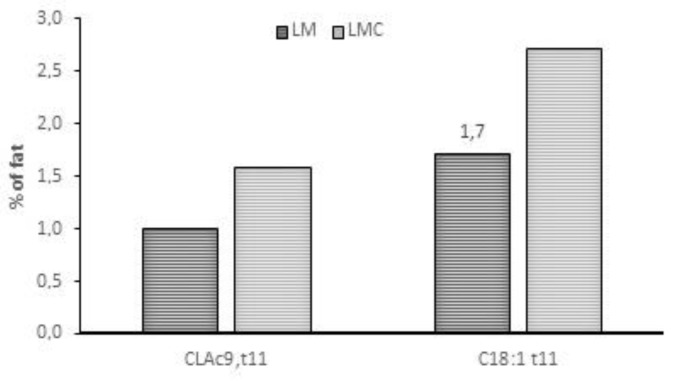
Level of CLA *cis*-9, *trans*-11 and vaccenic acid in milk from sheep fed with binary mixture (ryegrass + burr medic, LM) compared to ternary mixture (ryegrass + burr medic + *Chrysanthemum coronarium*, LMC), [42].

**Figure 8 animals-09-00771-f008:**
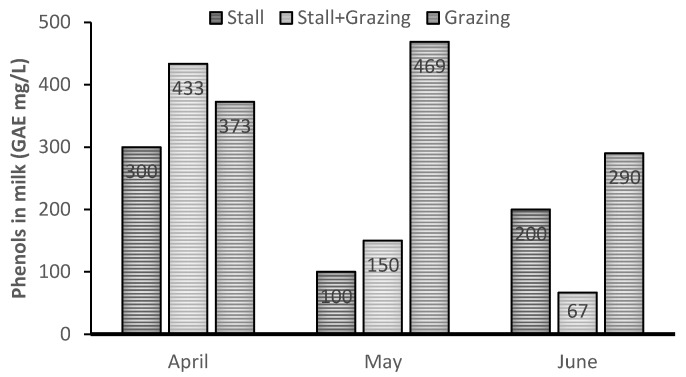
Effect of different goat feeding systems on milk phenols content [47].

**Figure 9 animals-09-00771-f009:**
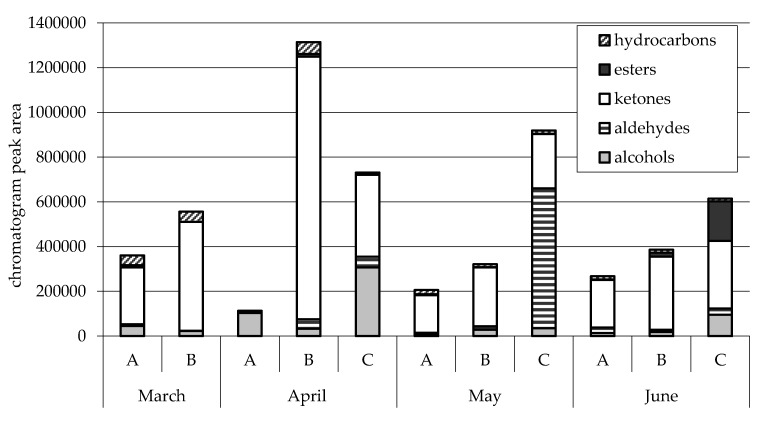
Effect of different feeding goat system (A = stall-fed with hay and concentrate; B = woody and herbaceous species with small amount of supplements; C = only herbaceous species with small amount of supplements) on volatile compounds [50].

**Table 1 animals-09-00771-t001:** Fatty acid classes and indices of milk from sheep grazing only fresh herbage compared to different levels of oilseed supplementation [44].

		Treatment				SEM	*p*-Value
		PAS	NFS	SLNA	SALA		
MUFA	% FAME	24.67 ^ab^	22.27 ^b^	28.62 ^a^	28.81 ^a^	0.89	0.04
PUFA	% FAME	6.68 ^c^	6.89 ^bc^	9.84 ^a^	9.58 ^ab^	0.34	0.01
SFA/UFA	ratio	2.14 ^ab^	2.44 ^a^	1.66 ^b^	1.66 ^ab^	0.09	0.03
AI (index)	ratio	2.06 ^ab^	2.63 ^a^	1.87 ^b^	1.88 ^ab^	0.10	0.02
n-3/n-6	ratio	2.08 ^a^	0.71 ^bc^	0.60 ^c^	0.90 ^b^	0.12	0.01
I-Harris	% FAME	0.13	0.10	0.10	0.12	0.01	0.06
UTFA	% FAME	0.30 ^b^	0.35 ^b^	0.60 ^a^	0.54 ^a^	0.03	0.01
UTFA/CLA	ratio	0.13 ^b^	0.19 ^a^	0.23 ^a^	0.19 ^a^	0.01	0.01

PAS = only pasture (24 h); NFS = pasture (3 h) + non fat enriched supplement; SLNA = pasture (3 h) + fat enriched supplement (based on 10.60% of raw sunflower seed and 4.80% of linseed on DM basis, respectively); SALA = pasture (3 h) + fat enriched supplement (based on 10.50% of linseed and 2.90% of raw sunflower seed on DM basis respectively); I-Harris = (DHA + EPA) SEM = standard error of means; UTFA = (C18:1 t9+C18:2 t9, t12); UTFA/CLA = (UTFA/CLAc9t11)*100; FAME = fatty acid methyl ester; different letters between columns represent the effect of treatment (^a,b^ Differ at least at *p* < 0.05).

**Table 2 animals-09-00771-t002:** Levels of vitamin A, vitamin E, cholesterol (mg/100 g fat) and degree of antioxidant protection (DAP) in milk and cheese from goats reared with different feeding systems [46].

		Vit. A	Vit. E	Cholesterol	DAP(*1000)
Milk	C	0.82a	0.68a	407a	1.67
	HH	0.97ab	3.66b	314b	11.66
	LH	1.09b	3.52b	294b	11.97
Cheese 24 h	C	0.68a	0.58a	394a	1.47
	HH	0.92b	3.74b	316b	11.84
	LH	1.06c	3.76b	309b	12.17
Cheese 60 days	C	0.66a	0.65a	390a	1.67
	HH	0.86b	3.74b	320b	11.69
	LH	1.01c	3.76b	304b	12.37

C = Stall feeding; HH = High herbage cover shrubland; LH = Low herbage cover shrubland; DAP = degree of antioxidant protection.

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
