# Peer review of "Extensive Ruminant Production Systems and Milk Quality with Emphasis on Unsaturated Fatty Acids, Volatile Compounds, Antioxidant Protection Degree and Phenol Content"

_animals, 2019, doi:10.3390/ani9100771_

Round 1

Reviewer 1 Report

This is an excellent review paper regarding the beneficial effects of dairy products derived from grazing ruminants on human health. The authors make a holistic approach and successfully provide the currently available evidence. They also interestingly present statistical analyses that back up their conclusions. The manuscript is well written. There are two comments that have to be addressed, though, before the manuscript is accepted for publication.

I suggest that the authors should give more emphasis on the redox/antioxidant effects of grazing on the ruminant products and their relation to the improvement of human health. That is, they are requested to add two relevant sections in the discussion (one for cows and one for sheep/goat). Furthermore, if it is possible they can propose redox biomarkers at section 4 (Plant biomarkers). They could possibly find some relevant information in the following paper (doi: 10.1016/j.cotox.2018.10.001).

Although the figures are professionally built, the authors should improve the quality.

Author Response

Reviewer 1

This is an excellent review paper regarding the beneficial effects of dairy products derived from grazing ruminants on human health. The authors make a holistic approach and successfully provide the currently available evidence. They also interestingly present statistical analyses that back up their conclusions. The manuscript is well written. There are two comments that have to be addressed, though, before the manuscript is accepted for publication.

AU: thanks

Comment 1

I suggest that the authors should give more emphasis on the redox/antioxidant effects of grazing on the ruminant products and their relation to the improvement of human health.

AU: done

That is, they are requested to add two relevant sections in the discussion (one for cows and one for sheep/goat).

AU: done

Furthermore, if it is possible they can propose redox biomarkers at section 4 (Plant biomarkers).

AU: done

They could possibly find some relevant information in the following paper (doi: 10.1016/j.cotox.2018.10.001).

AU: Thanks for the help and suggestion to improve the quality of paper

Comment

Although the figures are professionally built, the authors should improve the quality. 

AU: we improve the quality of pictures 1, 3 , 4, 5, 6, 7. We hope that is ok now

Reviewer 2 Report

The topic of the review of the research undertaken by the authors is very interesting and current. It touched upon the subject of the positive impact of the extensive feeding system of ruminants on health quality and the technological characteristics of milk and dairy products. An interesting aspect of this work is the possibility of using pasture markers to track and authenticate ruminant products.

The paper requires minor revision and systematisation of the presented information.

The title is not completely adequate to the content of the article. The manuscript is mainly focused on grazing diets of ruminants and the positive effect of grazing on the quality of milk and dairy products. But in addition, the possibility of using pasture markers to track and authenticate ruminant products has been characterized. Therefore, it should also be reflected in the title.

Detailed comments:

The figures in the manuscript are of low quality and illegible. Often without appropriate explanations.

Page 2 - Figure 1 is illegible and it is mainly related to the font size and color.

Page 3 - Figure 3 has no legend or explanations. What is hz, hx, hy ......?

L85 „many authors (....)”?

L85-86 Please expand the relationship between edaphic and flora characteristics and the aroma of dairy products.

L90 – “…. the fatty acid profile of milk and cheese (Figure 3).” In this publication, the authors did not present results of analysis of cheeses.

Page 7, L208 - OBCFA abbreviation not explained

L209 correct mistakes "antitannic sipplementation."

L300-301 the next number for this section is rather 2.3.

Author Response

Reviewer 2

The topic of the review of the research undertaken by the authors is very interesting and current. It touched upon the subject of the positive impact of the extensive feeding system of ruminants on health quality and the technological characteristics of milk and dairy products. An interesting aspect of this work is the possibility of using pasture markers to track and authenticate ruminant products.

AU: thanks

The paper requires minor revision and systematisation of the presented information.

AU: done see below

The title is not completely adequate to the content of the article. The manuscript is mainly focused on grazing diets of ruminants and the positive effect of grazing on the quality of milk and dairy products. But in addition, the possibility of using pasture markers to track and authenticate ruminant products has been characterized. Therefore, it should also be reflected in the title.

AU: since the authentication procedure is not very discussed along the text and because the title is too long we prefer do not change the title 

Detailed comments:

The figures in the manuscript are of low quality and illegible. Often without appropriate explanations.

AU: see below the comments

Page 2 - Figure 1 is illegible and it is mainly related to the font size and color.

AU: to be done

Page 3 - Figure 3 has no legend or explanations. What is hz, hx, hy ......?

AU: done

L85 „many authors (....)”?

AU: done

L85-86 Please expand the relationship between edaphic and flora characteristics and the aroma of dairy products.

AU: done

L90 – “…. the fatty acid profile of milk (Figure 3).” In this publication, the authors did not present results of analysis of cheeses.

AU: done we deleted cheese

Page 7, L208 - OBCFA abbreviation not explained

AU: done

L209 correct mistakes "antitannic sipplementation."

AU: done

L300-301 the next number for this section is rather 2.3.

AU: done

Reviewer 3 Report

The authors reviewed literature related to the effects of pasture' agronomic composition on the dietary characteristics of milk and dairy products. It is not particularly innovative the study on the fatty acid profile. On the other hand, the chapters where are reported the effects of the botanical essences on organoleptic characteristics of dairy products seem more interesting.

Author Response

Reviewer 3

The authors reviewed literature related to the effects of pasture' agronomic composition on the dietary characteristics of milk and dairy products. It is not particularly innovative the study on the fatty acid profile. On the other hand, the chapters where are reported the effects of the botanical essences on organoleptic characteristics of dairy products seem more interesting.

L 67: AU: done we add Mancinelli reference

L85: AU: done, we delete …..

L97: AU:done we add D’urso reference

L116: AU: done we insert “I”

L120: AU: done deleted

L124: AU: done

L213: AU: done

L322 : AU: done

L324: AU: done